# Ten-year follow-up of auditory brainstem implants: From intra-operative electrical auditory brainstem responses to perceptual results

Sheila Veronese[1]*, Marco Cambiaghi[1], Nicola Tommasi[2], Andrea Sbarbati[1], John J. Galvin, III[3]

1 Department of Neuroscience, Biomedicine and Movement Sciences, Verona University, Verona, Italy, 2 Centre of Economic Documentation (CIDE), Verona University, Verona, Italy, 3 House Institute Foundation, Los Angeles, California, United States of America

* sheila.veronese@univr.it

**Data Availability Statement:** All relevant data are within the paper and its Supporting Information files.

## Abstract

The auditory brainstem implant (ABI) can provide hearing sensation to individuals where the auditory nerve is damaged. However, patient outcomes with the ABI are typically much poorer than those for cochlear implant recipients. A major limitation to ABI outcomes is the number of implanted electrodes that can produce auditory responses to electric stimulation. One of the greatest challenges in ABI surgery is the intraoperative positioning of the electrode paddle, which must fit snugly within the cochlear nucleus complex. While presently there is no optimal procedure for intraoperative electrode positioning, intraoperative assessments may provide useful information regarding viable electrodes that may be included in patients' clinical speech processors. Currently, there is limited knowledge regarding the relationship between intraoperative data and post-operative outcomes. Furthermore, the relationship between initial ABI stimulation with and long-term perceptual outcomes is unknown. In this retrospective study, we reviewed intraoperative electrophysiological data from 24 ABI patients (16 adults and 8 children) obtained with two stimulation approaches that differed in terms of neural recruitment. The interoperative electrophysiological recordings were used to estimate the number of viable electrodes and were compared to the number of activated electrodes at initial clinical fitting. Regardless of the stimulation approach, the intraoperative estimate of viable electrodes greatly overestimated the number of active electrodes in the clinical map. The number of active electrodes was associated with long-term perceptual outcomes. Among patients with 10-year follow-up, at least 11/21 active electrodes were needed to support good word detection and closed-set recognition and 14/21 electrodes to support good open-set word and sentence recognition. Perceptual outcomes were better for children than for adults, despite a lower number of active electrodes.

**Funding:** The author(s) received no specific funding for this work.

**Competing interests:** The authors have declared that no competing interests exist.

## Introduction

The multichannel auditory brainstem implant (ABI) is a surgically implanted neuro-prosthetic device developed to electrically stimulate auditory neurons of the cochlear nucleus complex (CNC) bypassing the auditory nerve. It is used to restore hearing sensation in patients for whom a cochlear implant (CI) is not effective and/or applicable. Initially, it was indicated for patients affected by neurofibromatosis type 2, who were totally deaf after acoustic neuroma removal [1]. Over time, its indications have been extended to adults with other non-tumor diseases [2–4] and children with cochlear nerve aplasia [4] or severe inner ear malformations [5]; however, indications remain controversial for pathologies such as neuropathy and trauma with temporal bone fracture [6, 7].

While the ABI can provide hearing to patients in whom auditory nerve function is impaired, perceptual outcomes are often poorer than those for cochlear implant recipients, in whom the auditory nerve remains functional [8, 9]. For most patients, the benefit of the ABI is restricted to sound awareness, partial identification of ambient sounds, or as aid for lip-reading. Only a limited number of ABI patients achieve the ability to recognize speech without using lip-reading [10]. These relatively poor outcomes may be because the ABI is mainly indicated for adult patients with tumors, in whom the neural tissue is compromised by the presence of the tumor and/or by tumor resection [11, 12]. Non-tumor patients, such as children with cochlear nerve aplasia, patients with ossification or malformation of the cochlea, and patients with profound hearing loss after head trauma with cochlear fractures, usually perform better after receiving an ABI [13, 14].

One factor that affects ABI outcomes is the coupling of the electrode paddle with the CNC during surgery. Indeed, effective electrode placement is essential to provide patients with auditory sensation while avoiding stimulation of surrounding non-auditory anatomical structures [15]. While electrode position has been strongly correlated with ABI speech recognition ability, other aspects may play a crucial role in perceptual outcomes, and it is unclear which aspects are most prominent [16]. [part moved to discussion].

In the present study, we evaluated interoperative electrophysiology and 10-year follow-up data from adult and pediatric ABI patients to investigate whether: 1) the morphology of intraoperative electrophysiology used to guide electrode paddle positioning differs between two stimulation protocols, 2) intraoperative electrophysiology might be used to predict the number of electrodes activated after surgery and, 3) the number of electrodes at initial activation is associated with auditory outcomes over the long term.

## Materials and methods

Surgical and electrophysiological procedures were approved by the Ethics Committee of Verona Hospital. Written informed consent was obtained from the adult patients and or from the parents/caregiver of pediatric patients. This study was carried out in accordance with the Declaration of Helsinki. Note that all data presented in this study were collected as standard of care for the ABI recipients.

### Patient population

A retrospective case series analysis was performed to review data from 24 patients who received Cochlear Nucleus ABIs (Cochlear Ltd., Sydney, Australia) at the ENT Department in Verona between June 2004 and September 2007. Sixteen patients were adults (8 females, 8 males) and 8 were children (5 females, 3 males). At the time of surgery, adult patients were aged 21 to 59 years (mean = 37.69 ± 13.65) and children were aged 1.42 to 10.25 years (mean = 4.09 ± 2.84).

Inclusion criteria were 10 years of follow-up, the ability to communicate orally in Italian for adults, and the ability of family members to report on communication for children unable to speak. Exclusion criteria were the presence of motor deficits or body malformations that prevented perceptual testing. Note that children with mental delay were not excluded from the study because their diagnosis was made later in years based on other learning delays.

## Surgical procedure

A retrosigmoid approach was used for ABI implantation [17–20]. After electrode paddle insertion and before closure, electrically-evoked auditory brainstem response (EABR) measurements were made to optimize ABI electrode placement. The evoked potentials were selected because they were more appropriate than other cortical potentials in terms of the presence and stability of responses [21], and because they were indifferent to anesthesia [22, 23].

When an appropriate positioning of the electrode paddle over the CNC was obtained, the implant was stabilized with suturing before surgery conclusion [16, 17]. During all surgical procedures, facial and lower cranial nerves were monitored to detect unwanted non-auditory stimulation.

## Intraoperative EABR

For EABR recordings, the Amplaid MK12 electrodiagnostic system (Amplifon SpA, Milan, Italy) was used. Recording settings and parameters are detailed in Veronese et al. [24]. Patients were tested with two different stimulation protocols: one suggested by Cochlear Ltd (CP) [25, 26], and a modified protocol (MP) [24]. The main difference between the two protocols is the distance between the active and return electrodes, which is smaller for the MP. The motivation for the MP was to reduce the number of neural fibers recruited by stimulation, which generally increases with distance between the active and return electrodes due to channel interaction and electrical current spread [24].

EABR waveforms were analyzed according to Waring [22, 27–30] in terms of the number of peaks [22, 27], latencies [22, 27–30], and amplitudes [29]. Different numbers of waveforms were recorded for each patient. If electrode paddle repositioning was required to optimize implant placement, tests were repeated. The data included in the present study were from the final recordings, with the paddle in its final position.

## ABI activation procedure

ABIs were activated 4–6 weeks after surgery, based on patient recovery. Adult activation took place in intensive care units with cardiac and respiratory monitoring, in direct collaboration with patients who were asked to report any auditory and non-auditory sensations or any psycho-physical alterations. Threshold and comfort levels of each electrode were defined with a down-up-down procedure. Current levels, quantized by Cochlear as current units (CUs), were progressively increased until the patient reported an auditory response, defined as threshold. To identify the comfort level, CUs were further increased until the patient reported a discomfortable perception (e.g., too intense or unpleasant). After these initial estimates of threshold and comfort levels, current levels were reduced in 1-CU steps. If non-auditory sensations were reported, the electrode was excluded from the initial speech processor map.

In the pediatric population, postoperative EABRs were recorded before activation to guide the initial speech processor map. Recordings were performed under sedation with cardiac and respiratory monitoring.

The same intraoperative equipment and electrode montages were used for EABR recordings. EABRs were evoked using common ground stimulation mode, in which current is

delivered to the target electrode and all the other electrodes are used as the ground/return electrodes. The pulse phase duration was 150 μs, the stimulation rate was 25 pulses per second (pps), and current was decreased from 190 CUs to the hearing threshold level in 10-CU steps. Test and retest recordings were performed to identify auditory responses.

EABRs were interpreted as follows. Electrodes presenting non-auditory components (peak latency > 4–4.5 ms) or unclear/poorly defined responses were excluded from the activation map. After EABR recording and while the children were waking up in a separate room, initial maps were created based on the identified thresholds. Before activation, the initial stimulation levels were decreased to be below the EABR thresholds and then gradually increased in 5-CU steps while observing the child's behavioral responses.

## Long-term perceptual outcomes

Ten years after the activation of the implant, perceptual abilities were assessed [31]. Testing was performed in a quiet room, with the examiner orally producing the stimuli (words or sentences, depending on the test). The examiner was seated 1 meter away from the seated participant, in a latero-posterior position and ipsilateral to the implant. As such, the participant could not see the examiner, but excellent sound propagation was guaranteed without any attenuation. If the participant had any residual acoustic hearing, adequate masking was performed by administering white noise [32]. Perceptual abilities were categorized in terms of levels of performance:

- Level 0: no sound awareness of sounds and words presented at 65 dBA. Here, participants needed only to indicate that they heard a sound.

- Level 1: ≥60% correct detection of sounds and words presented at 65 dBA. Here also, participants needed only to indicate that they heard a sound.

- Level 2: ≥60% closed-set disyllabic word identification. An *n*-alternative forced choice was used (3AFC, 5AFC, or 10AFC, depending on participants' age and/or appropriate level of difficulty). Disyllable words were used because, unlike English language, there are few monosyllabic words in Italian language. Participants were presented with a test word (e.g., caf-fe, boc-ca, tu-ta, to-po, ri-so), and had to choose among the response choices (e.g., "caffe", "tuta", and "riso" for the 3AFC task). The response choices were shown on a sheet of paper (images for children, text for adults). Ten test runs words for the 3AFC, 5AFC, and 10AFC tasks, resulting in a total of 30, 50 or 100 words tested for each participant.

- Level 3: ≥60% open-set word and sentence recognition. Disyllable words (different from those used in the previous tests) and simple everyday sentences were used. During testing, the examiner presented the stimulus (word or sentence), and the participant repeated as accurately as possible. The examiner scored the number of words and words in sentences correctly identified

All tests were administered with appropriate levels of difficulty according to age at testing and cognitive level.

## Data analysis

For the EABRs, Pearson's chi-squared test was used to compare the distribution of peaks recorded with the two stimulation protocols. Student's t-test was used to compare differences in peak characteristics across the protocols. Through analysis of intraoperative EABRs, the number of active electrodes was hypothesized for both protocols and compared to the number of active electrodes included in patients' clinical maps.

The number of patients that obtained the different perceptual levels at 10-year follow-up was expressed in terms of percentage. The mean and range for the number of active electrodes was calculated across patients for Level 1 (sounds and words detection), Level 2 (closed-set word recognition), and Level 3 (open-set word and sentence recognition). The number of active electrodes and perceptual results were compared using a Probit model (0 = no results, awareness, and detection; 1 = identification, recognition, and comprehension). The median and 95% confidence interval (percentile) of the perceptual results, correlated with the number of active electrodes, were calculated.

## Results

### Patient characteristics

Fig 1 shows a STROBE (Strengthening the Reporting of Observational Studies in Epidemiology) flow chart that illustrates the patient selection process.

Table 1 shows demographic information for the patients included in the study. All adults were born with acoustic hearing, except for AD1. Seven out of 16 adults (43.75%) presented with a tumor disease, and the ABI was placed during tumor removal surgery. All adults had progressive hearing loss related to the progress of their pathological states, except for AD6 and AD11. Ten out of 16 adults used hearing aids for at least 5 years before receiving the ABI; AD9 successfully used a contralateral CI for 9 years before receiving the ABI. The remaining adults did not use hearing aids before receiving the ABI (AD11 due to sudden hearing loss; the others because the hearing aids did not provide sufficient gain). At the time of ABI surgery, all adults were diagnosed with severe-to-profound hearing loss. All children were diagnosed with severe-to-profound hearing loss before the age of one, except for CH3, who was diagnosed after the age of five. CH1, CH2 and CH3 did not use hearing aids before receiving the ABI. The remaining children used hearing aids for at least 6 months before receiving the ABI, but with no improvement in speech perception or production. In addition to hearing aids, CH4 used a CI for 1.42 years before ABI, without any benefit.

### Waveform morphology

EABRs were recorded in 13 patients with the CP, 4 with the MP, and 7 with both protocols. Thus, EABRs were recorded in 20 patients with the CP and 11 with the MP. Overall, 896 waveforms were recorded (379 with CP, and 517 with MP) (S1 Dataset).

Examples of EABRs obtained with the two protocols are shown in Figs 2 and 3. The number of peaks in the waveforms ranged from 1 to 3. There was no significant difference in the distribution of waveform peaks between the two protocols ($\chi^2[3]$ = 6.1136; p = 0.106). MP resulted in a reduction in the number of 0- and 1-peak waveforms, and an increase in the number of 2- and 3-peak waveforms (Table 2).

For both protocols, P2 was the most frequently observed peak, and P1 was the least frequently observed peak. Although there was a greater percentage of evident P2 and P3 with MP than with CP, the difference was not significant (P1: p = 0.7630; P2: p = 0.3321; P3: p = 0.1382). The mean peak amplitude across peaks was greater with CP than with MP. The difference in peak amplitude across the protocols was significant for P2 (p = 0.0499), but not for P1 (p = 0.5792) or P3 (p = 0.1663). There was no significant difference in mean amplitude between protocols for P1 (p = 0.3299), P2 (p = 0.5828, or P3 (p = 0.1622).

### Number of electrodes during surgery and at activation

The mean number of viable electrodes based on intraoperative EABR analysis and the mean number of active electrodes in the clinical map are shown in Table 3. There was little difference

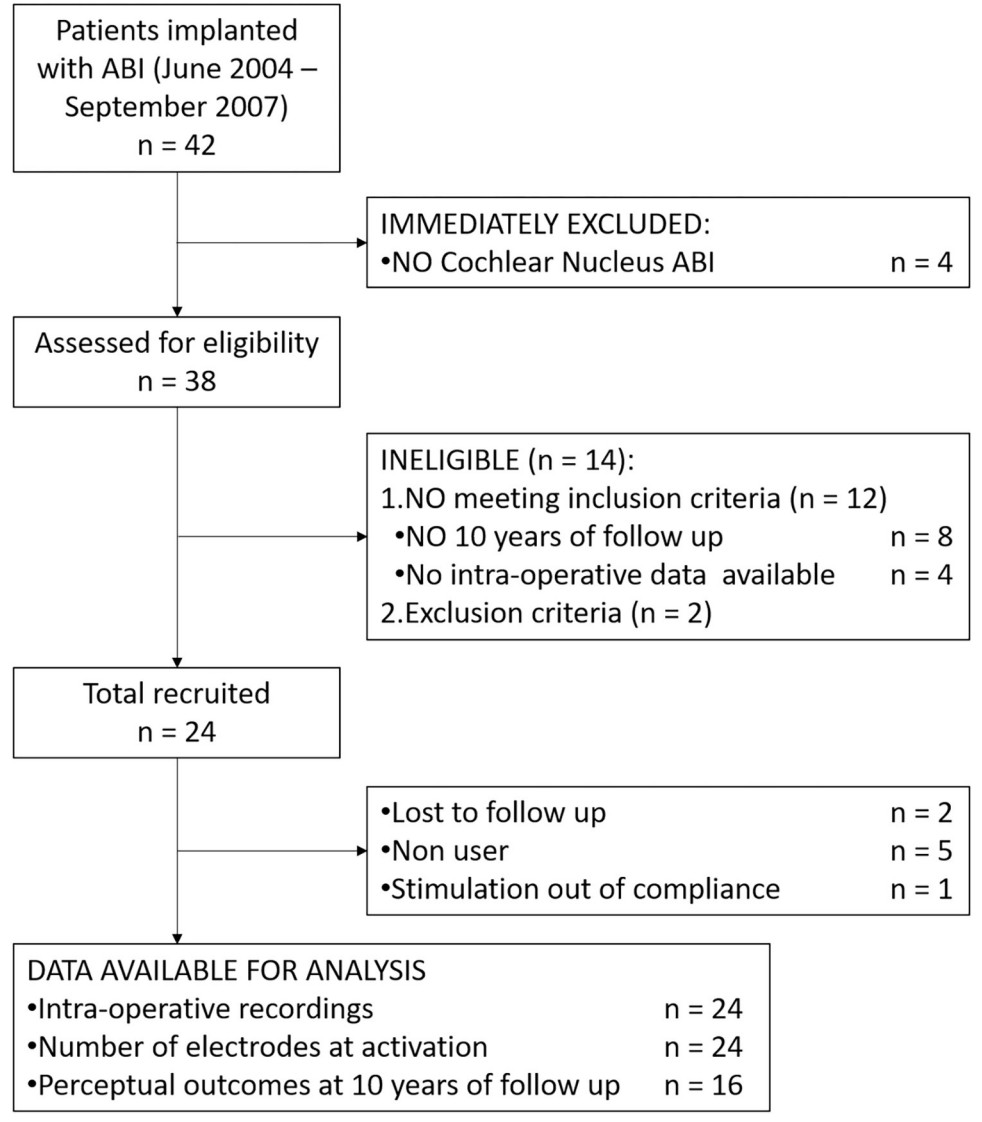

**Fig 1. STROBE flow diagram of patient selection.**

between CP and MP in terms of the mean number of viable electrodes from EABRs. For the patients where both protocols were used to record EABRs (AD9, AD10, AD11, AD13, AD15, AD16, CH7), a paired t-test showed no significant different in EABRs between the two protocols [t(6) = -0.2, p = 0.850]. However, the number of viable electrodes from EABRs (mean = 18.9±2.5) greatly overestimated the number of electrodes in the clinical map (mean = 13.5±4.6. A paired t-test was performed on the electrode data to compare the number of viable electrodes from EABRs to the number of active electrodes in the clinical map; EABR data were averaged across the protocols for the patients in whom both CP and MP were used. The number of viable electrodes from EABRs was significantly higher than the number of active electrodes in the clinical map [t(30) = 6.9, p < 0.001]. The difference between the number of viable electrodes from EABRs and active electrodes in the clinical map was ≤ 2 in 20.8% of patients, > 2 ≤ 4 in 16.7% of patients, > 4 ≤ 6 in 16.7% of patients, and > 6 in 45.8% of patients.

**Table 1. Patient demographic information.**

| Patient | Sex | Age at ABI (years) | ABI side | Etiology | Contralateral hearing at 10 years follow-up | Hearing aid use | |
|---|---|---|---|---|---|---|---|
| | | | | | | Before ABI | At 10 yrs follow-up |
| AD1 | F | 25 | Right | Bilateral Mondini malformation | N | Y | N |
| AD2 | M | 56 | Right | NF2 | Profound hearing loss | Y (C) | Y (C) |
| AD3 | M | 46 | Right | NF2 | N | N | N |
| AD4 | F | 21 | Right | Ossification (post-meningitis) | N | Y (B) | N |
| AD5 | F | 58 | Left | Neuropathy | Profound hearing loss | N | N |
| AD6 | M | 41 | Right | Trauma with monolateral temporal fractures; contralaterally post-meningitis ossification | N | Y | N |
| AD7 | F | 28 | Right | NF2 | N | Y | N |
| AD8 | F | 22 | Right | NF2 | Profound hearing loss | Y (C) | Y (C) |
| AD9 | M | 52 | Left | Ossification (result of an otosclerotic process) | Profound hearing loss | Y | N |
| AD10 | F | 34 | Left | Ossification (and Ménière's disease) | N | N | N |
| AD11 | M | 36 | Right | Trauma with monolateral temporal fractures; contralaterally previous acoustic neuroma exeresis surgery | N | N | N |
| AD12 | M | 22 | Left | NF2 | N | N | N |
| AD13 | F | 42 | Right | Neuropathy | N | Y | N |
| AD14 | F | 59 | Right | Acoustic neuroma | Severe hearing loss | Y (C) | Y (C) |
| AD15 | M | 39 | Right | Neuropathy | N | N | N |
| AD16 | M | 22 | Left | NF2 | N | N | N |
| CH1 | F | 2.17 | Right | Cochlear malformation | N | N | N |
| CH2 | F | 5.5 | Right | Aplasia (and bilateral Mondini malformation) | N | N | N |
| CH3 | F | 10.25 | Left | Hypoplasia | N | N | N |
| CH4 | M | 4.92 | Left | Aplasia | N | Y | N |
| CH5 | M | 3.33 | Right | Aplasia (and Goldenhar syndrome) | N | Y | |
| CH6 | M | 2.5 | Right | Cochlear malformation | N | Y | N |
| CH7 | F | 1.42 | Right | Hypoplasia (and bilateral Mondini malformation) | N | Y | N |
| CH8 | F | 2.67 | Right | Aplasia | N | Y | N |

ABI, auditory brainstem implant; AD, adult; CH, child; CI, cochlear implant; F, female; M, male; NF2, neurofibromatosis type 2; Y, yes; N, no; C, contralateral; B, bilateral

## Perceptual results

Perceptual data and the number of active electrodes in the clinical map are shown in Table 4. Two patients (8.3%, one adult and one child) were lost to follow-up. Five patients (20.8%) did not use the ABI. Two of these patients used a contralateral CI or hearing aid. Three of these patients stopped using the ABI due to poorer than expected perceptual outcomes. Interestingly, one patient continued to use the ABI despite no sound awareness. Another patient stopped using the ABI because electrical stimulation levels became out of compliance (i.e., the maximum current delivered by the ABI was insufficient to provide any extra-auditory or auditory sensation).

Only 62.5% of patients (9 adults and 6 children) obtained substantial benefit from the ABI. Of these 15 patients, 26.6% (4 adults) reached Level 1, 33.3% (3 adults and 2 children) reached Level 2, and 42.8% (2 adults and 4 children) reached Level 3. Four of the 9 adults were tumor patients (44.4%) and 5 were non-tumor patients (55.56%). Among the 4 tumor patients, 2 reached Level 1, one reached Level 2, and another reached Level 3. Among the 5 non-tumor patients, 2 reached Level 1, 2 reached Level 2, and one reached Level 3.

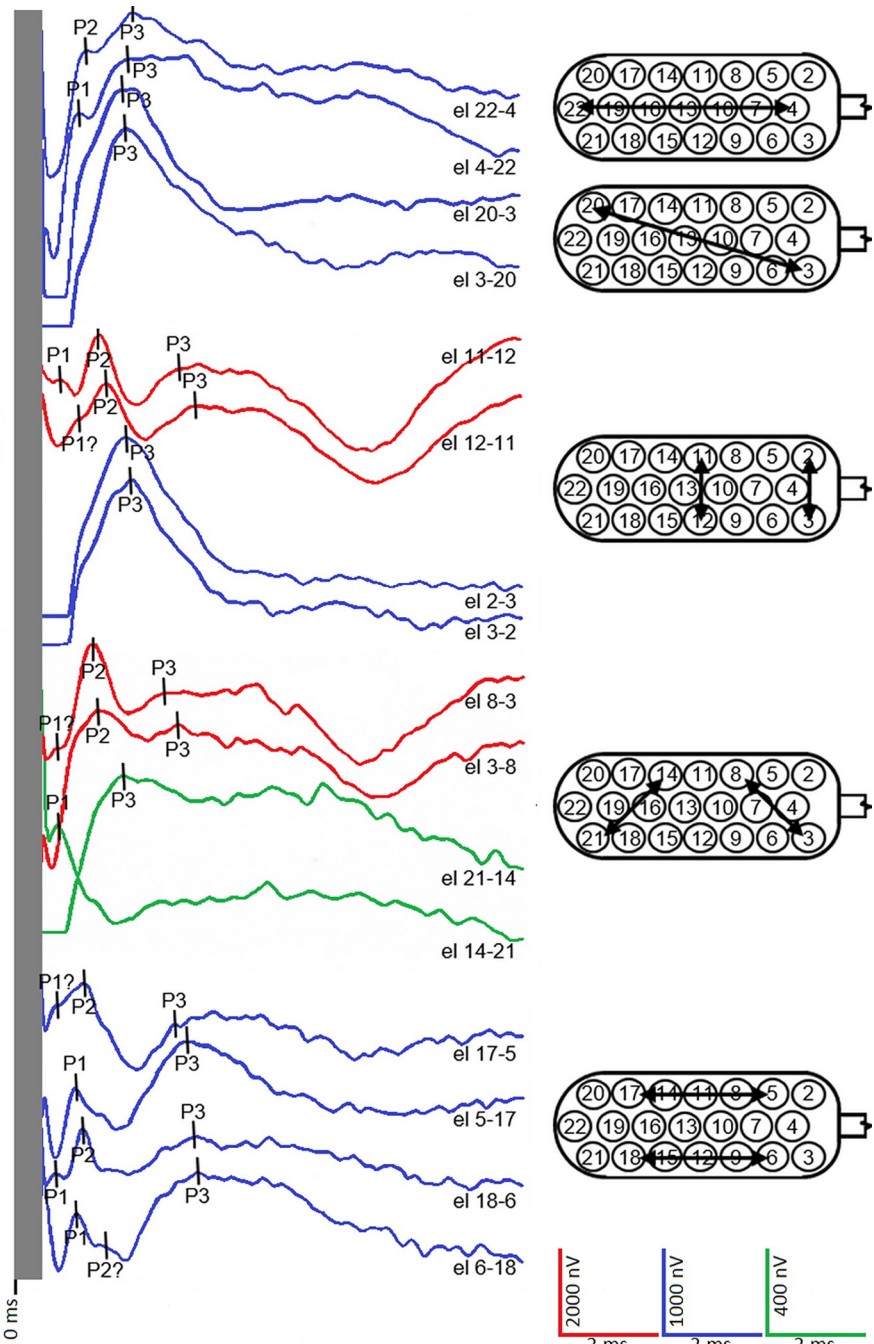

**Fig 2. EABRs obtained with the cochlear protocol (CP) for patient CH7.** For electrode combinations with wide stimulation arcs (e.g., 22–4, 3–20), the waveforms have a 1- or 2-peak morphology more frequently than a 3-peak morphology. For electrode combinations with narrower stimulation arcs (e.g., 11–12, 8–3), the waveforms have a 3-peak morphology when associated with auditory sensation. Stimulation artifacts present in the first 0.5 ms of recording have been eliminated. el, electrodes.

## Perceptual results versus the number of active electrodes

Table 4 shows the number of active electrodes in the clinical map, perceptual performance, and perceptual level (see above) for the adult and pediatric patients. Perceptual performance

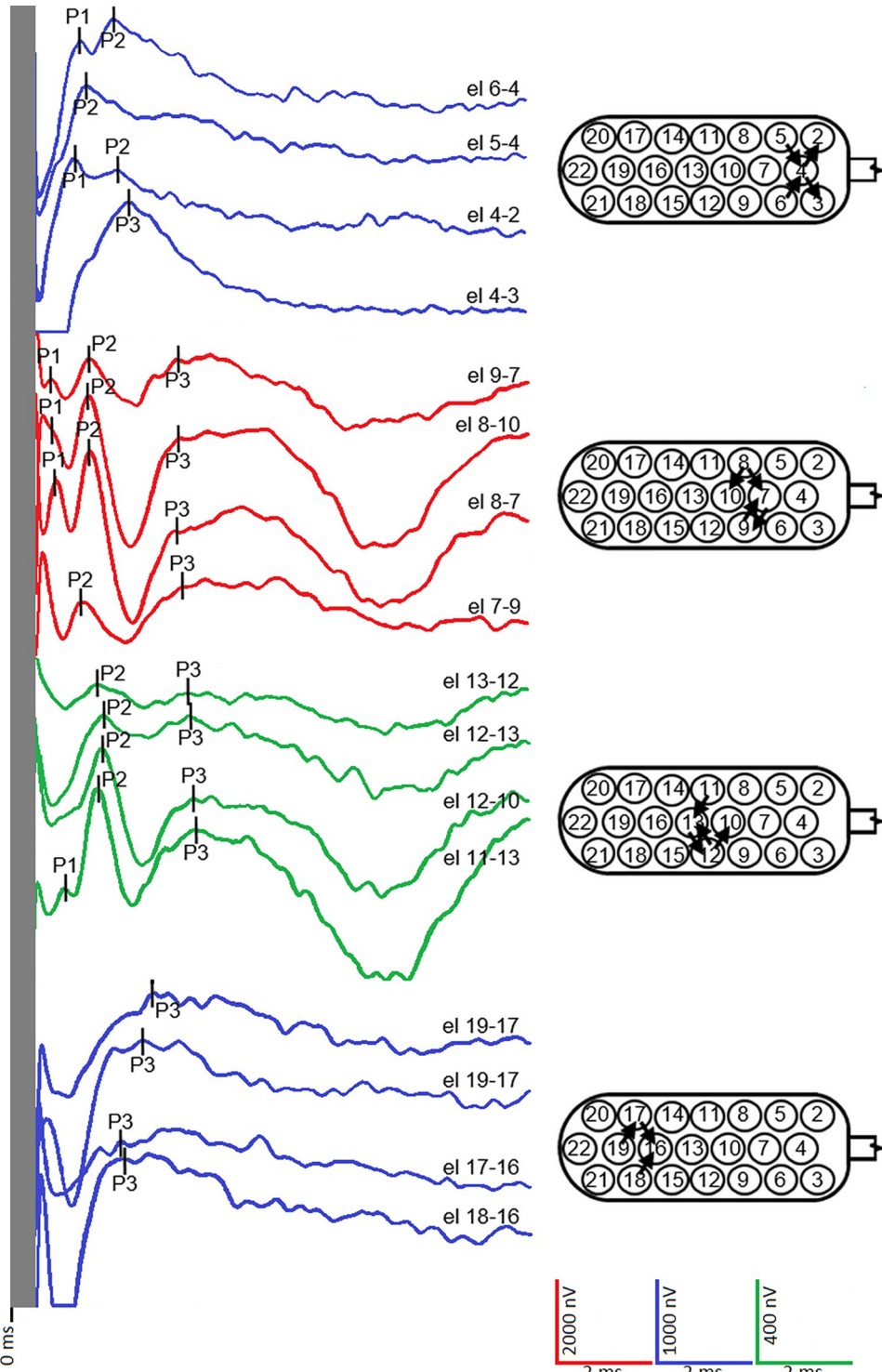

**Fig 3. EABRs obtained with the modified protocol (MP) for patient CH7.** The electrode combinations have narrower stimulation arcs compared to CP (Fig 2) by reducing the electric field and interactions between electrodes, which allows for better interpretation of electrode position. In general, there are a greater number of 3-peak waveforms than with CP (Fig 2). Stimulation artifacts present in the first 0.5 ms of recording have been eliminated. el, electrodes.

**Table 2. Peaks characterization of the EABR waveforms recorded with the two different stimulation protocols.**

| Stimulation protocol | | Number of peaks | | | | Presence of peak | | | Peak amplitude (nV) | | |
|---|---|---|---|---|---|---|---|---|---|---|---|
| | | 0 | 1 | 2 | 3 | P1 | P2 | P3 | P1 | P2 | P3 |
| CP | Number | 29 | 106 | 178 | 66 | 145 | 276 | 239 | 292 | 473 | 407 |
| | Percentage | 7.7 | 28.0 | 47.0 | 17.4 | 38.3 | 72.8 | 63.1 | | | |
| MP | Number | 28 | 119 | 259 | 111 | 182 | 409 | 381 | 193 | 408 | 274 |
| | Percentage | 5.4 | 23.0 | 50.1 | 21.5 | 35.2 | 79.1 | 73.7 | | | |

CP, Cochlear protocol; MP, modified protocol

was compared to the number of active electrodes in 16 patients who achieved Level 0, 1, 2, or 3. Three patients for whom data was unavailable (12.5%; 2 adults and 1 child) and five patients who did not use the ABI (20.8%; 4 adults and 1 child) were excluded from the analyses. Among the included patients, 1 patient (4%; 1 adult) reached Level 0, with 7 active electrodes; 4 patients (16.7%%; 4 adults) reached Level 1, with an average of 13.3±1.7 active electrodes (range = 11–15); 5 patients (20.8%; 3 adults and 2 children) reached Level 2, with an average of 15.8±4.1 active electrodes (range = 11–20); 6 patients (25%; 2 adults and 4 children) reached Level 3, with an average of 15.5±3.7 active electrodes (range = 10–20).

Fig 4A and 4B show the median probability for word detection and closed-set recognition (Levels 1–2; n = 10) and open-set word/sentence recognition (Level 3; n = 6) as a function of the number of active electrodes. The median probability of reaching Level 2 ranged from 19.0% with 7 active electrodes to 95.8% with 20 active electrodes. The median probability of reaching Level 3 ranged from 18.7% with 7 active electrodes to 53.1% with 20 active electrodes. Probit regression analysis between the number of active electrodes and perceptual results showed that at least 11 AEs were required to reach Levels 1–2 ($p \leq 0.01$), and at least 14 active electrodes were needed to reach Level 3 ($p = 0.005$).

Among adults, 4/9 (44%) achieved Level 1 with 11–15 AEs, 3/9 achieved Level 2 (33%) with 12–20 AEs, and 2/9 (22%) achieved Level 3 with 19 and 20 AEs. Among children, 0/7 (0%) achieved Level 1, 3/7 achieved Level 2 (43%) with 8–17 AEs, and 4/9 achieved Level 3 (57%) with 10–16 AEs. Thus, children performed better and needed less AEs to achieve both Level 2 and Level 3.

## Discussion

There are relatively few studies that have followed ABI patients for 10 years [13, 33]. As such, the present adult and pediatric 10-year follow-up data with the ABI are valuable, especially given the perceptual results. The present data also show an association between the number of active electrodes in the clinical map and perceptual results, and that perceptual outcomes were generally better for children than for adults.

**Table 3. Mean, standard deviation (Std), minimum (Min), and maximum (Max) number of viable electrodes estimated during surgery and at ABI activation.**

| | | CP | | | | MP | | | | Both | | | |
|---|---|---|---|---|---|---|---|---|---|---|---|---|---|
| | Patient | Mean | Std | Min | Max | Mean | Std | Min | Max | Mean | Std | Min | Max |
| Viable electrodes from surgery | Adult | 18.7 | 3.3 | 11 | 21 | 19.8 | 1.4 | 17 | 21 | 19.0 | 2.8 | 11 | 21 |
| | Child | 18.8 | 1.7 | 17 | 21 | 18.7 | 1.2 | 18 | 20 | 18.8 | 1.5 | 17 | 21 |
| | All | 18.6 | 2.9 | 11 | 21 | 19.5 | 1.4 | 17 | 21 | 18.9 | 2.5 | 11 | 21 |
| Active electrodes in MAP | Adult | 13.2 | 4.7 | 7 | 20 | 14.4 | 5.6 | 7 | 20 | 13.7 | 5.0 | 7 | 20 |
| | Child | 12.3 | 3.6 | 8 | 17 | 14.3 | 3.1 | 11 | 17 | 13.0 | 3.4 | 8 | 17 |
| | All | 13.0 | 4.4 | 7 | 20 | 14.4 | 4.9 | 7 | 20 | 13.5 | 4.6 | 7 | 20 |

**Table 4. Number of viable electrodes estimated from EABRS and at ABI activation, as well as perceptual results.**

| Patient | Number of active electrodes in clinical map | Perceptual results | Level | Notes |
|---|---|---|---|---|
| AD1 | 12 | 100% word identification (3AFC) | 2 | |
| AD2 | 13 | detection 30–60 dB HL | 1 | tinnitus |
| AD3 | 7 | no sound detection | n/a | |
| AD4 | 9 | out of compliance | n/a | |
| AD5 | 15 | detection 50–65 dB HL | 1 | |
| AD6 | 20 | 100% word identification (3AFC) | 2 | |
| AD7 | 14 | detection 30–60 dB HL | 1 | |
| AD8 | 19 | 100% word identification (3AFC) | 2 | |
| AD9 | 7 | detection 30–60 dB HL | n/a | non-user, contralateral CI |
| AD10 | 20 | 80% open-set word and sentence comprehension | 3 | |
| AD11 | 11 | detection 30–60 dB HL | 1 | |
| AD12 | 19 | 90% open-set word and sentence comprehension | 3 | |
| AD13 | 7 | detection 50–65 dB HL | n/a | non-user |
| AD14 | 19 | detection 50–65 dB HL | n/a | non-user, contralateral HA |
| AD15 | 13 | no sound detection | n/a | non-user |
| AD16 | 19 | lost in follow up | n/a | |
| CH1 | 10 | 100% open-set word and sentence comprehension | 3 | mental delay |
| CH2 | 13 | 100% open-set word and sentence comprehension | 3 | |
| CH3 | 8 | 100% word identification (3AFC) | 2 | non-user |
| CH4 | 16 | 100% words and sentences comprehension | 3 | |
| CH5 | 10 | lost in follow up | n/a | |
| CH6 | 15 | 100% open-set word and sentence comprehension | 3 | |
| CH7 | 17 | 100% word identification (3AFC) | 2 | mental delay |
| CH8 | 11 | 100% word identification (10 AFC) | 2 | |

AD, adult; CH, child; dB HL, hearing level in decibels; 3AFC, 3-alternative, forced choice; 10AFC,10-alternative, forced choice; CI, cochlear implant; HA, hearing aid

On average children had fewer active electrodes (12.5±3.3) in the clinical map than did adults (14.0±4.9), yet had better perceptual outcomes. This advantage may be due to may an uncompromised neural substrate [33], as the present children study did not present with

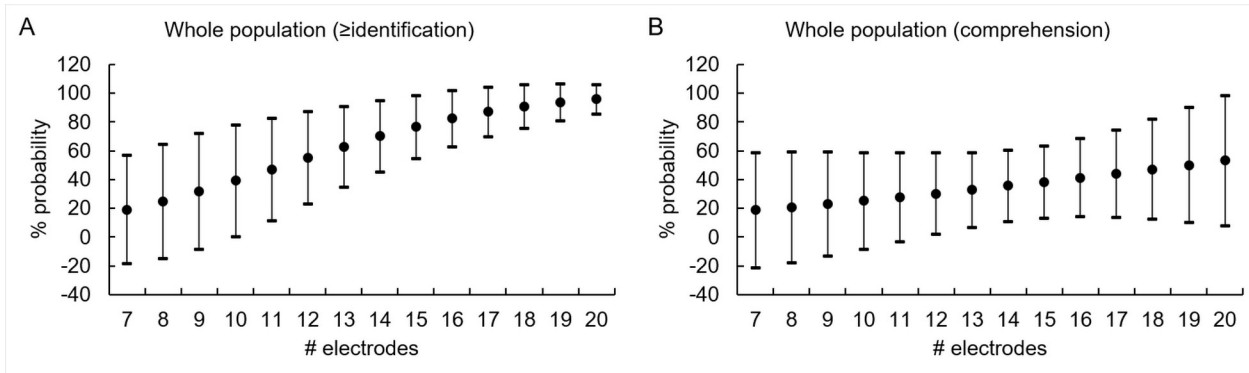

**Fig 4. Median percentage and 95% confidence interval of perceptual results.** A: Probability of closed-set word identification (Level 2) as a function of the number of active electrodes, across all ABI patients. The percentages increase non-linearly and the 95% confidence intervals narrow as the number of active electrodes increases. B: Probability of open-set word and sentence recognition (Level 3) as a function of the number of active electrodes, across all ABI patients. While the median percentage increases slowly with the number of active electrodes, the 95% confidence interval remains similarly wide across the number of electrodes, indicating considerable inter-subject variability. #, number.

tumors or neurodegenerative diseases. It is also possible that the greater neural plasticity in children may have also contributed to the better outcomes than observed in adults [34].

Consistent with previous studies, substantial variability in perceptual outcomes was observed among the present ABI patients; several hypotheses have been proposed to explain the large variability in perceptual outcomes among ABI patients [10, 16]. After surgery, the survival of the CNC cells is crucial, as they support modulation sensitivity, which has been significantly associated with speech perception [12, 13]. Speech recognition appears to be related to duration of deafness, electrode position, and the number of electrodes that produce distinct pitch percepts [16]. Perceptual outcomes have also been associated with lower levels of electric stimulation [35]. Better coupling of the ABI to the CNC results in lower stimulation current levels at initial activation [24], which may reduce non-auditory side effects and provide better perceptual results. Another factor that may contribute to the variability in ABI outcomes is that there are no standardized procedures regarding surgical approach, electrode positioning, the stimulation protocol used for recording intraoperative potentials, rehabilitation methods, and auditory evaluation tests, with patient age being a key factor. Finally, long-term perceptual data in ABI patients are scarce [13, 33], making it difficult to know which of these many factors contribute to long-term ABI outcomes.

The two intraoperative stimulation protocols did not produce significantly different EABRs. However, Veronese et al. [24] showed significant advantages for MP over CP in terms of saturation effects and electrical artifacts; MP better predicted most stimulation current levels at activation, even though the number of activated electrodes was the same between CP and MP. This suggests that it is possible to position the implant correctly with both protocols, but that evaluations carried out via MP better reflect the coupling of the ABI with the CNC.

Across all patients, the mean number of active electrodes during initial clinical fitting (13.5 ±4.4) was similar to values reported in previous studies [15, 20, 36, 38]. Note that the present study included a greater number of ABI patients than in these previous studies. With either stimulation protocol, the number of estimated viable electrodes from interoperative EABRs (18.8±2.7) greatly over-predicted the number of active electrodes at initial clinical fitting. One reason for this discrepancy may be related to the electrode paddle displacement. Wong et al. [23] observed differences between intra- and post-operative EABRs in pediatric ABI patients and suggested that the electrode paddle may shift between surgery and initial activations. However, Anwar et al. [38] suggested that the greater stability of post-operative EABRs, compared to the intraoperative recordings, may better reflect the stability of the position of the ABI. Because the position of the implant is stabilized during surgery [16], the change in the relative position of the CNC and the ABI electrode paddle could be attributed to the return to of the anatomical structures to their original position after being retracted during surgery. Furthermore, the variation in the position of the patient's head, from supine during surgery to upright after surgery, could cause a rostro caudal shift of both the anatomical structures and the electrode paddle. Behr et al. [16] stressed that the positioning of the electrode paddle is a key point, and that particular attention should be paid to the closure procedure; hazardous patient movements should be avoided during the first days after surgery.

The main problem of electrode shifting is the possible negative effect on perceptual results [16]. Considering the dimensions of the ABI paddle and the CNC [33, 35, 37–41] and the estimated position of the paddle [1, 22, 42, 43], even a minimal shift could cause undesired non-auditory stimulation [1]. This also relates to how many electrodes are needed to obtain perceptual results that are comparable to those with CIs. Behr et al. [16] found no significant correlation between the number of active electrodes in the clinical map and perceptual results. In that study, the best ABI performers had more than 66.7% of electrodes activated in their clinical maps (8/12 electrodes with the Med-El ABI). Kuchta et al. [44] suggested that 75% of

electrodes needed to be activated to obtain good perceptual results in users of the Cochlear Nucleus 22 ABI. These percentages are consistent with those of the present study.

One limitation to the present study is that the perceptual outcome categories were somewhat broad, enough so that they could be used to characterize both pediatric and adult perception. The categories represented increasing levels of difficulty, similar to the principles of the speech recognition hierarchy introduced by Geers (1994) and developed for the pediatric population [45]. As deafness in the adult ABI patients was due to tumors, neuropathy, head trauma, etc. (but not presbycusis), we felt that the present categories could be applicable to adults and children. While developmental differences between adults and children were not considered, children generally outperformed adults, possibly due to a better neural substrate [33], or greater neural plasticity [34]. Greater standardization of perceptual outcomes appropriate for adults and children would allow for better characterization of ABI outcomes.

## Conclusions

Different intraoperative stimulation protocols may result in morphological differences in the recorded EABRs. These differences were ultimately unrelated to the number of activated electrodes during initial clinical fitting. In fact, intraoperative EABRs greatly overpredicted the number of active electrodes in the clinical map.

However, there appeared to be a relationship between the number of active electrodes and long-term perceptual outcomes, with a greater number of electrodes needed for good performance. During the initial period of ABI stimulation, it is essential to have a large number of active electrodes to provide adequate acoustic neural stimulation. We found that, at 10 years of follow-up, a minimum of 11 active electrodes (52% of implanted electrodes) were needed to support good word detection and closed set recognition, and a minimum of 14 active electrodes (67% of implanted electrodes) were needed to support good open set word and sentence recognition. Interestingly, children needed fewer active electrodes as did adults to support the same perceptual performance.

Further standardization of all procedures related to the ABI is necessary, from intraoperative monitoring to clinical fitting to perceptual outcome measures. Further studies are underway to analyze the effects of age and etiology on intraoperative EABRs, and their relationship with long-term perceptual outcomes.

## Supporting information

**S1 Dataset.**
(XLS)

**S1 File. Multinomial logistic regression models.**
(DOCX)

## Acknowledgments

The authors thank Dr. Sebastiano Mininni for his help in EABR recording, all the speech therapists who evaluated the patients over the years (in particular, Dr. Alessia Sandini), and Mr. Stefano Setti for his graphic support. But our most heartfelt thanks are to Steve Otto, a friend, a maestro.

## Author Contributions

**Conceptualization:** Sheila Veronese, Andrea Sbarbati.

**Data curation:** Sheila Veronese.

**Formal analysis:** Sheila Veronese, Nicola Tommasi, John J. Galvin, III.

**Investigation:** Sheila Veronese.

**Methodology:** Sheila Veronese.

**Project administration:** Andrea Sbarbati.

**Validation:** Marco Cambiaghi, Andrea Sbarbati.

**Visualization:** Sheila Veronese, Marco Cambiaghi.

**Writing – original draft:** Sheila Veronese, Marco Cambiaghi.

**Writing – review & editing:** Sheila Veronese, Marco Cambiaghi, Andrea Sbarbati, John J. Galvin, III.

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
