## [Decision Letter · Decision Letter 0]

27 Dec 2022

PONE-D-22-23874Ten-year follow-up of auditory brainstem implants: From intra-operative electrical auditory brainstem responses to perceptual resultsPLOS ONE

Dear Dr. Veronese,

Thank you for submitting your manuscript to PLOS ONE. After careful consideration, we feel that it has merit but does not fully meet PLOS ONE’s publication criteria as it currently stands. Therefore, we invite you to submit a revised version of the manuscript that addresses the points raised during the review process.

We look forward to receiving your revised manuscript.

Kind regards,

Paul Hinckley Delano, Ph.D.

Academic Editor

PLOS ONE

Journal Requirements:

Reviewers' comments:

Reviewer's Responses to Questions

**Comments to the Author**

1. Is the manuscript technically sound, and do the data support the conclusions?

Reviewer #1: Yes

Reviewer #2: Yes

2. Has the statistical analysis been performed appropriately and rigorously? 

Reviewer #1: No

Reviewer #2: Yes

3. Have the authors made all data underlying the findings in their manuscript fully available?

Reviewer #1: Yes

Reviewer #2: Yes

4. Is the manuscript presented in an intelligible fashion and written in standard English?

Reviewer #1: Yes

Reviewer #2: Yes

5. Review Comments to the Author

Reviewer #1: This is a retrospective cohort study aimed at assessing whether intraoperative electrophysiological data from 24 ABI patients correlated with postoperative perceptual outcomes.

The abstract is adequate, and has listed rationale and setting details, alongside the most important findings.

The introduction section is too long, and several paragraphs discussing details regarding hearing outcomes should be transferred to the discussion section.

The objectives of the study are presented clearly and the introduction section communicates the need for investigating the electrophysiological variables impacting perceptual outcomes of ABI surgery. I would also recommend using a STROBE flowchart and checklist. Inclusion and exclusion criteria need to be clearly stated. I would consider including children with mental delay a significant confounding factor, but due to the small number of patients, exclusion is not neccessary, but rather an ancillary statistical analysis with mental delay as a separate binary variable to control sampling bias.

However, since the methodology included a small number of patients, evaluated through tests assesing correlation, but not causality. I would suggest using an ancillary multivariate multinomial logistic regression model if a causal connection is to be evaluated between the three groups of perceptual outcomes (level 0-3) and number of electrodes.

The tables and figures are adequate and support data interpretation. I have no concerns regarding internal or external manuscript validity. The discussion is fine, addressing all of the objectives of the manuscript, alongside other published relevant studies.

A honest limitations and shortcomings paragraph is needed.

Reviewer #2: I enjoyed reading this article. It presents innovative and new data to the research community regarding ABI. As this is a rare surgery, performed in a few centers worldwide, it is of great interest to publish long term results for ABI patients. The retrospective study is well designed; very well written article and interesting findings. Good figures and tables.

Only two little minor details and can be published:

page 7: The evoked potentials were selected because more adequate than other potentials in terms of... missing a word

page 10: eliminate the period before Level 3 (line 189)

6. PLOS authors have the option to publish the peer review history of their article (what does this mean?). If published, this will include your full peer review and any attached files.

Reviewer #1: No

Reviewer #2: No

---

## [Author Response · Author response to Decision Letter 0]

25 Jan 2023

To the editor:

We thank the editor and the reviewers for their helpful comments. We have incorporated nearly all suggestions into the revised MS. Below we respond to specific comments. Please let us know if you need further information.

Sincerely,

Sheila Veronese

1. Is the manuscript technically sound, and do the data support the conclusions?

Reviewer #1: Yes

Reviewer #2: Yes

2. Has the statistical analysis been performed appropriately and rigorously? 

Reviewer #1: No

Reviewer #2: Yes

3. Have the authors made all data underlying the findings in their manuscript fully available?

Reviewer #1: Yes

Reviewer #2: Yes

4. Is the manuscript presented in an intelligible fashion and written in standard English?

Reviewer #1: Yes

Reviewer #2: Yes

5. Review Comments to the Author

Reviewer #1: This is a retrospective cohort study aimed at assessing whether intraoperative electrophysiological data from 24 ABI patients correlated with postoperative perceptual outcomes.

The abstract is adequate, and has listed rationale and setting details, alongside the most important findings.

>>Thank you.

The introduction section is too long, and several paragraphs discussing details regarding hearing outcomes should be transferred to the discussion section.

>>We have moved some of the Introduction to the Discussion. For the inclusion criteria, we now state: “Inclusion criteria were 10 years of follow-up, the ability to communicate orally in Italian for adults, and the ability of family members to report on communication for children unable to speak. Exclusion criteria were the presence of motor deficits or body malformations that prevented perceptual testing. Note that children with mental delay were not excluded from the study because their diagnosis was made later in years based on other learning delays.”

The objectives of the study are presented clearly and the introduction section communicates the need for investigating the electrophysiological variables impacting perceptual outcomes of ABI surgery. I would also recommend using a STROBE flowchart and checklist. 

>> We have added a new figure (Figure 1) which presents a STROBE flow diagram of the patient selection process. 

Inclusion and exclusion criteria need to be clearly stated.

>>In the original MS (and unchanged here), we explicitly stated the inclusion and exclusion criteria. 

I would consider including children with mental delay a significant confounding factor, but due to the small number of patients, exclusion is not necessary, but rather an ancillary statistical analysis with mental delay as a separate binary variable to control sampling bias.

However, since the methodology included a small number of patients, evaluated through tests assessing correlation, but not causality. I would suggest using an ancillary multivariate multinomial logistic regression model if a causal connection is to be evaluated between the three groups of perceptual outcomes (level 0-3) and number of electrodes.

>> Two multivariate logistic regression models were evaluated, one with 15 subjects (excluding the single subject with score 0) and one with 13 subjects (also excluding the 2 mentally delayed children) - file attached as Supplemental Informations. Although neither model detected a significant relationship between the number of active electrodes and 10 yr outcome data, there was a positive trend, consistent with the results previously reported in the MS. Accordingly, we prefer to retain the previous analysis. If the reviewer and editor think it’s worthwhile, we could present the multivariate logistic regression models as supplementary data.

The tables and figures are adequate and support data interpretation. I have no concerns regarding internal or external manuscript validity. The discussion is fine, addressing all of the objectives of the manuscript, alongside other published relevant studies.

>>Thank you.

An honest limitations and shortcomings paragraph is needed.

>>We have added to the end of the Discussion: “One limitation to the present study is that the perceptual outcome categories were somewhat broad, enough so that they could be used to characterize both pediatric and adult perception. The categories represented increasing levels of difficulty, similar to the principles of the speech recognition hierarchy introduced by Geers (1994) and developed for the pediatric population [45]. As deafness in the adult ABI patients was due to tumors, neuropathy, head trauma, etc. (but not presbycusis), we felt that the present categories could be applicable to adults and children. While developmental differences between adults and children were not considered, children generally outperformed adults, possibly due to a better neural substrate [33], or greater neural plasticity [34]. Greater standardization of perceptual outcomes appropriate for adults and children would allow for better characterization of ABI outcomes.”

Reviewer #2: I enjoyed reading this article. It presents innovative and new data to the research community regarding ABI. As this is a rare surgery, performed in a few centers worldwide, it is of great interest to publish long term results for ABI patients. The retrospective study is well designed; very well written article and interesting findings. Good figures and tables.

>>Thank you.

Only two little minor details and can be published:

page 7: The evoked potentials were selected because more adequate than other potentials in terms of... missing a word

>>Revised as: “The evoked potentials were selected because they were more appropriate than other cortical potentials in terms of the presence and stability of responses [21], and because they were indifferent to anesthesia [22,23].

page 10: eliminate the period before Level 3 (line 189)

>>Corrected.

---

## [Editor Report · Decision Letter 1]

13 Feb 2023

Ten-year follow-up of auditory brainstem implants: From intra-operative electrical auditory brainstem responses to perceptual results

PONE-D-22-23874R1

Dear Dr. Veronese,

We’re pleased to inform you that your manuscript has been judged scientifically suitable for publication and will be formally accepted for publication once it meets all outstanding technical requirements.

Kind regards,

Paul Hinckley Delano, Ph.D.

Academic Editor

PLOS ONE
---

## [Editor Report · Acceptance letter]

22 Feb 2023

PONE-D-22-23874R1 

Ten-year follow-up of auditory brainstem implants: From intra-operative electrical auditory brainstem responses to perceptual results 

Dear Dr. Veronese:

I'm pleased to inform you that your manuscript has been deemed suitable for publication in PLOS ONE. Congratulations! Your manuscript is now with our production department. 

Kind regards, 

on behalf of

Dr. Paul Hinckley Delano 

Academic Editor

PLOS ONE